# CREATIVE ROBOT TOOL USE BY COUNTERFACTUAL REASONING

## ABSTRACT

We propose a causal reasoning framework for creative robot tool use in which a novel object is correctly identified to functionally substitute for a tool that is no longer available to the robot. During training, our framework discovers properties of the source tool that are causally relevant to the task by conducting counterfactual experiments in a physics-based dynamics model. We reconstruct the geometry of the tool from vision and systematically intervene on its properties using 3D shape editing to generate counterfactual tool variants. During deployment, candidate tools are classified based on their similarity to the source tool with respect to the discovered causal features. By reconstructing the task in a dynamics model, our approach grounds tool use in the physics of the problem. We illustrate our approach in reaching a distant object with different sticks, in scooping candies from a bowl with different kitchen items, and in using different boxes as platforms to step up to retrieve an object from a high shelf. We surpass state-of-the-art methods in repurposing everyday objects and show that the discovered features align with human judgment.

## 1 INTRODUCTION

Creative tool use is considered an essential trait of intelligence (Call, 2013). Nature provides inspiring examples such as crows that drop stones to increase water level (Jelbert et al., 2014), and apes that stack crates and use sticks to retrieve a high-hanging banana (Köhler, 2018). Humans can exhibit more complex and creative tool use even at an early age (Guerin et al., 2013), but emulating the same behavior in robotics remains challenging (Fitzgerald et al., 2021). Although recent advances improved robot capability to perform complex, contact-rich tasks (Kroemer et al., 2021), creative tool use, which is defined as using tools beyond their intended purpose, is an open problem.

Traditionally, robot tool use has been explored through the lens of affordances—the action possibilities an object offers given an agent's capabilities (Gibson, 1979; Şahin et al., 2007). Prior work has focused on learning visual features (Mar et al., 2017), object categories (Sinapov & Stoytchev, 2007), and keypoints (Turpin et al., 2021) to predict affordances, but these methods often struggle to generalize due to the limited diversity of robot-collected interaction data. Recent approaches leverage large vision-language models (VLMs) trained on web-scale multimodal data to inject commonsense reasoning into tool selection (Ahn et al., 2022; Driess et al., 2023). However, such models often lack grounding in the robot's embodiment and physical environment, leading to failures when proposed tools are physically infeasible or unsuitable for the task.

We propose a causal reasoning framework for creative tool use that enables a robot to identify and repurpose a substitute tool for a task beyond its original intent. Our approach **generalizes** to novel objects, is **grounded** in physical constraints, identifies **causal** features, and provides **human-interpretable** justifications for its decisions. Our key contribution is to integrate the common sense and generalization capabilities of Visual Language Models (VLM) with the physical fidelity of a dynamics model to provide a principled approach to causal feature discovery for tool selection and policy transfer for tool manipulation. During the *training* phase, our approach assumes that the robot can do the task with a *source* object with a previously acquired skill. It first reconstructs the given task scene in a physics-based simulator using vision-based mesh reconstruction and performs the task with counterfactual objects created by a semantic mesh editor (Ganeshan et al., 2024) whose physical properties can be edited in the simulator. In doing so, the robot counterfactually discovers

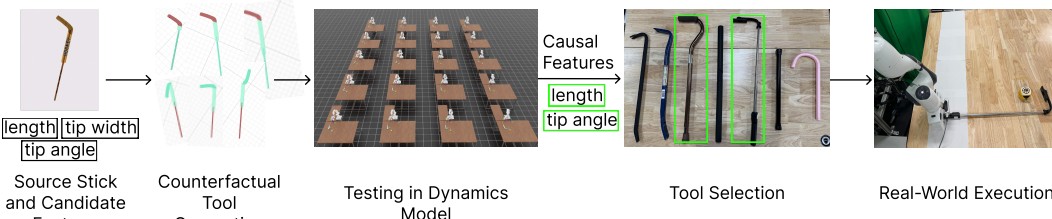

Figure 1: An overview of the pipeline. For a given source object and the task definition, a VLM proposes a set of object features that might affect the task success. Using the candidate features as axes of variation, a 3D semantic shape editing tool generates counterfactual objects, and by carrying out the task in simulation, the robot discovers the causal features which then can be used to find out substitutes for creative robot tool use.

the properties of the tool that are both causally relevant to the task and satisfy the physical limitations of the robot. We leverage VLMs to suggest candidate features for the tool, making our approach suitable for open-world domains. During *deployment*, the robot edits the source object to make it similar to *target* objects (unmodeled objects available in the novel environment) for all inferred causal features. If the edited source object is predicted to be successful through simulation execution, that object is suggested for real robot execution. By identifying and reasoning with causally relevant features, our approach is robust to distractors and can readily explain why certain objects were not suitable for the task. Experiments in three real world scenarios from table-top and mobile manipulation show that the proposed framework correctly identifies causal features and finds alternative tools to complete tasks.[1]

## 2 BACKGROUND AND RELATED WORK

We are interested in one-shot tool manipulation problems. We assume that the robot previously acquired the skill to complete the task with a source tool, and our goal is to identify a novel tool that is a functional substitution for the source tool among many unmodeled objects to complete the task. After successful tool selection, the skill might be adapted to the new task for successful execution. We studied rigid object interactions in pulling, scooping tasks for table-top manipulation and reaching tasks in mobile manipulation. We review the related tool use works under the taxonomy of Qin et al. (2023). Here, we are introducing the most related works. For a more extensive related work section, please see A.2.

Part-level affordances that encode the action possibilities are leveraged to enable transfer between intercategorical tools (Myers et al., 2015; Schoeler & Wörgötter, 2016; Kroemer et al., 2012). Fitzgerald et al. (2019; 2021) transferred tooltip pose constraints via human correction trajectories, while Agostini et al. (2015) used an affordance knowledge base for tool substitution in salad-making. We similarly use VLMs as a knowledge base but ground the output in simulation.

End-to-end methods predicted actions from visual features for sweeping and hammering (Fang et al., 2018; Xie et al., 2019). Task-specific keypoint predictors with procedural tool generation improved intercategory use in pushing, reaching, and hammering, using task info as environment keypoints (Qin et al., 2020) or rewards (Turpin et al., 2021). Procedural methods (Qin et al., 2020; Turpin et al., 2021; Fang et al., 2018) generated X, L, and T-shaped tools by combining convex parts, requiring 600, 10k, and 18k respectively training tools. By contrast, we use LLM-driven semantic generation, avoiding large-scale training and directly encoding affordances. Combining global and local geometry (Liu et al., 2024; Qin et al., 2021) transfers contact points but still requires demonstrations and ignores physical constraints. Zhu et al. (2015) focused on predefined physical features, which we find essential for tool use.

VLMs have recently been applied to reasoning tasks using affordances. Vision work (Tang et al., 2023; Huang et al., 2024; Yu et al., 2024) aligns text encodings of tool knowledge with visual task

---

[1]Extended results are available at `https://toolanalogies.github.io`. See a pedagogical example in Section A.1.

features for tool selection. However, such models lack task dynamics, causal reasoning, and robot interaction data. In robotics, VLMs have been used for tool selection and policies: Ren et al. (2023) trained meta-policies from tool descriptions; Lee et al. (2024) used LLMs as symbolic planners; and Car et al. (2024) combined high- and low-level planners but limited affordances to grasp prediction. Robotool (Xu et al., 2024), closest to our work, extracts concepts and plans with privileged knowledge of object layouts and properties. By contrast, we only reconstruct the source tool, infer or ignore physical properties, and satisfy constraints through simulation. Other approaches generate new tools via VLMs (Lin et al., 2025; Gao et al., 2025; Liu et al., 2023), whereas we repurpose everyday objects instead of designing or fabricating novel ones.

## 3 COUNTERFACTUAL REASONING FOR TOOL ADAPTATION

Consider a TV remote that falls under the sofa where you cannot reach directly. You might look for an object to help retrieve it. Intuitively, you can rule out a book for being too short, a crowbar for being too heavy, or a chair for being too large to fit under the gap. Instead, you might try a rolling pin or a selfie stick. But how do you know which object is suitable for the task? This judgment relies on understanding which physical properties, such as length, weight, or shape, are relevant for the task at hand, and whether a candidate object satisfies those properties. Humans can reason about this effortlessly using prior experience, commonsense knowledge, and an internal model of how physical interactions work. Please refer to A.1 to see the motivation in a toy grid environment

In this work, we show that a similar form of reasoning can be enabled in robots by combining commonsense priors from vision language models (VLMs) with counterfactual reasoning through simulation. We assume that the robot knows how to perform the task with a *source* tool. When placed in a new environment in which the source tool is absent, the robot must identify and use a *novel* (unmodeled) tool for the same task. Without a principled approach, it would spend a lot of time trying objects while violating task and robot constraints. We propose **ToolAnalogy**, a novel approach that discovers object properties that are causally related to task success by experimenting with counterfactual objects generated with a 3D semantic object editor, and uses these causal features to classify novel objects as substitutes to carry out the task.

### 3.1 PROBLEM FORMULATION

We formulate our problem by a Markov Decision Process (MDP) $M = \langle S, A, R, T, \gamma \rangle$ where $S$, $A$, $T$, $R$, $\gamma$ denote state space, action space, transition function, reward function, and discount factor, respectively. We assume that the state can be factored into objects $s = (o_a, o_1, o_2, ..., o_n)$ where $o_a$ denotes the agent and the others denote objects in the environment. The reward, defined by the task description $\mathcal{T}$, is a binary function for task success. We are interested in tasks in which the robot must use one of the objects as an intermediary tool ($o_{\text{tool}}$) to accomplish the task. We model each object using a feature vector consisting of semantic features, which could be exhaustively listed to cover all possible features $F$ and which are all known in common sense knowledge. For example, for a hockey stick, one can say rigid, has an angled head, long, thin, etc. However, it is impractical to assume that all can be listed or that this superset list can be used for classification. Instead, we discover a subset of causally relevant features that are sufficient to satisfy **the task at hand** $o_i = (x_1^i, x_2^i, , ..., x_k^i), X \subset F$. Note that the relevant features of the tool can change between tasks. For pulling, the angled head is very important, whereas it is irrelevant to be used as a paperweight.

### 3.2 METHOD

Figure 2 outlines our framework. First, given a task image and a description, a VLM (feature suggester) produces candidate 'make or break' tool features that are semantically related to the task at hand. Using these features as axes of variation, a semantic object editor generates a dataset of counterfactual objects. Then, the robot identifies the causal features by experimenting with these objects in simulation by reconstructing the real world task using a real-to-sim-to-real approach. Finally, by constructing a classifier with the causal features, the robot can figure out whether an unseen—unmodeled—tool can be used as a substitute for the task. While it is possible to verify the test object in simulation as an additional safety measure, our approach does not require any

simulation step once the causal classifier is constructed, which puts our method in a fundamentally new direction in creative robot tool use.

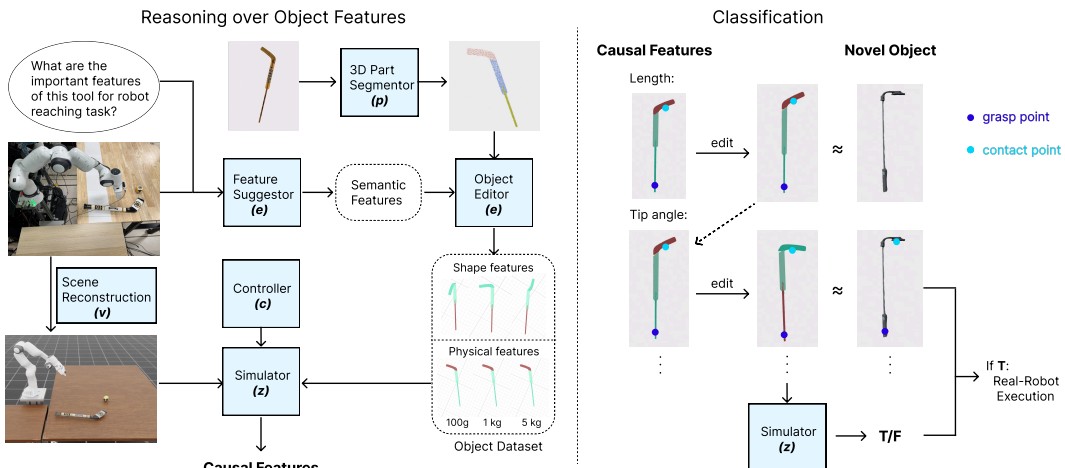

Figure 2: The tool selection pipeline before real-world execution. After finding out the causal features, the source object is morphed to match the dimensions of the target object using the semantic object editor and Chamfer distance as the metric.

To identify causally relevant features of a tool in the real world, we intervene on its features one at a time to generate modified versions of the tool (Pearl, 2009; Lee et al., 2021). The robot skill is then executed in simulation on the generated tools to understand the causal effect of interventions on task success. This procedure requires the following functions (annotated in blue boxes in Figure 2):

**Feature Suggester** ($\phi$). The object feature suggester that gets source object image $I_s$, task description $\mathcal{T}$ which includes implementation details if the controller and returns possible feature values, $\phi(I_s, \mathcal{T}) = \{x_1, x_2, \ldots, x_n\}$. We use ChatGPT-o3 (OpenAI, 2024) as the feature suggester in our experiments. We leverage VLMs to get semantic features of the source tool because learning causal variables from data is an important open problem (Schölkopf et al., 2021), and VLMs act as a good database for commonsense knowledge to provide candidate features. We run the VLM 10 times and get the top 6 most voted features out of 12. The features of the source stick for our task would be length, tip angle, tip length, tip thickness, grasp thickness, and mass.

**3D Part segmentor** ($p$). Here, we use a model that automatically identifies object parts without requiring a text query, as our shape editor, ParSEL, operates on object parts. The part names we are looking for are unknown and vary across different objects. SAMPART3D (Yang et al., 2024) satisfies this requirement. It takes a 3D model of the object, finds its parts with a granularity scale, and names them. This scale is a hyperparameter, and 1.5 works the best for our use case. For the hockey stick in the working example, we get the hockey handle, the hockey shaft, and the hockey blade as parts.

**Object Editor**($e$). The object editor that gets a feature $x_i$ and the 3D object model $O$ and returns the edited object model, $e(x_i, O) = O^*_{x_i}$. We use ParSEL (Ganeshan et al., 2024) to edit 3D objects using semantic shape features. It is based on program synthesis and gives reliable performance in comparison with other data-based methods. It takes object parts and an edit request as input and produces an edit program that can generate shapes with varying scales of that request to create a dataset. For physical features like mass, we edit physical properties in the simulator. See object dataset in Figure 2 where edits for tip angle and mass are illustrated.

**Scene Reconstruction** ($v$). Vision model to get 3D model of **only the source tool and goal objects** using images. We are not finding reconstructions of the candidate tools because this could again take a long time for an increasing number of objects. An object model with the same name could be downloaded from open datasets, and its dimensions can be scaled to match the real tool approximately, as an alternative to 3D reconstruction. Exact matching is not required unless the controller fails to use the tool. For reconstruction, we are using AR Code Code (2025) as suggested by a recent real-to-sim-to-real pipeline Torne et al. (2024), which creates the 3D model of the source object $O_s$

from its images $I_s$, $v(I_s) = O_s$. We assume the 3D models of a table for robotic arms and a floor for mobile manipulators. See scene reconstruction in Figure 2.

**Controller** ($c$). We use keypoint representations to define skills, following Liu et al. (2024), which maps a single demonstration trajectory from a source tool to candidate tools by matching keypoints using DINOv2 and local curvature features. We also assume a demonstration represented by keypoints. In our object dataset, keypoint transfer isn't required, transformations are handled automatically by the edit function. See that the grasp and contact points in Figure 3 (right) are naturally transformed during the edit of the parts where they are located. For a novel object, we locate the closest point to the keypoint on the most similar tool using Chamfer distance. The process of identifying the most similar tool is detailed in the Classification section.

**Dynamics Model** ($z$). We have access to an external dynamics model $z$ where we replicate the real world scene and a robot controller $c$ to achieve the task there. The model takes the edited object meshes $O^*_{x_i}$ and controller $c$, rolls out an episode and returns a boolean to report task success as specified in the task description, $z(\mathcal{T}, O^*_{x_i}, c) = \{0, 1\}$. We used the simulators IsaacSim (NVIDIA, 2021) with IsaacLab (Mittal et al., 2023) wrapper for table-top manipulation and MuJoCo (Todorov et al., 2012) for mobile manipulation. Figure 2 shows a scene from IsaacSim.

**Feature Classifier**($k$). The feature classifier checks if the novel (replacement) object has the causally relevant features $x$ found in the reasoning step. It uses RGBD image $I_r$ of the replacement object and the RGBD images of edited objects in the dataset $\mathcal{O}^* = \{O^*_{x_1}, \ldots, O^*_{x_n}\}$: $k(x, I_r, \mathcal{O}^*)$. This procedure is illustrated in Figure 2, right. We use the edit functions of each causal feature and make it as similar as possible to the novel object using the Chamfer distance. Figure 2 shows that the length source stick is scaled to match length of the novel object, 'the selfie stick'. Then, its tip angle is changed to be the same as the selfie stick. This process continues for all object features and repeated one more time to get the best match. The limitations of this process is discussed in the experiments section, therefore we also employ an additional classifier (VLM) to get a second opinion. This time it uses RGBD image $I_r$ of the replacement object and the RGBD image of boundary objects in the dataset. Here, boundary objects $O^*_{x_l}$, $O^*_{x_h}$ means the objects that successfully worked for the task with the lowest and the highest causal feature ($x$) values: $k(x, I_r, O^*_{x_l}, O^*_{x_h}) = \{0, 1\}$.

As our models utilize pre-trained foundational models, no training is necessary in any of the components apart from the skill acquisition procedure. In case the suggested tool does not satisfy the task, we employ a 2 step solution: First, we hypothesize that the all causal features have not been discovered, and we ask for additional feautures from the feature suggester and run the pipeline again. Second, we hypothesize that there are other unaccounted failures like sim-to-real gap, and we continue to try the next suggested tool by the pipeline.

## 4 EXPERIMENTAL RESULTS AND DISCUSSION

To evaluate the efficacy of our methodology, we perform experiments in three real-world table-top and mobile manipulation domains involving diverse objects and tool-use scenarios (Figure 3). The goal of our experiments is to study (1) the efficacy of our approach in reasoning about novel objects, (2) generalization across diverse tool-use scenarios, and (3) interpretability of the decisions made by our approach.

**Baselines.** We compare our approach with state-of-the-art approaches from the vision and robotics community that leverage VLMs and geometry heuristics for tool selection without requiring any training. (1) **ChatGPT-o3** (OpenAI, 2024) is a strong baseline that has been shown to be capable of performing complicated vision-language reasoning tasks. We provide ChatGPT-o3 with an image of all the available objects and prompt it to select the most relevant object for the task at hand. We also run two other variations (o3 RGB-D and o3 t-PCL) where we take RGB-D images of the scene and tools where source and each target tool are side by side, and transformed pointclouds to robot frame of source and target objects separately. (2) **CoTDet** (Tang et al., 2023) is a pretrained large vision model trained on the CoCo dataset (Lin et al., 2014), that can predict tool affordances with rationales to support the prediction. We provide an image of all the tools and then sort them based on confidence scores. (3) **MAGIC** (Liu et al., 2024) is a recent robotics approach for generalization of manipulation strategies to novel objects by contact-point matching using **DinoV2** features and local curvature analysis. We provide an image of each tool separately and compute confidence

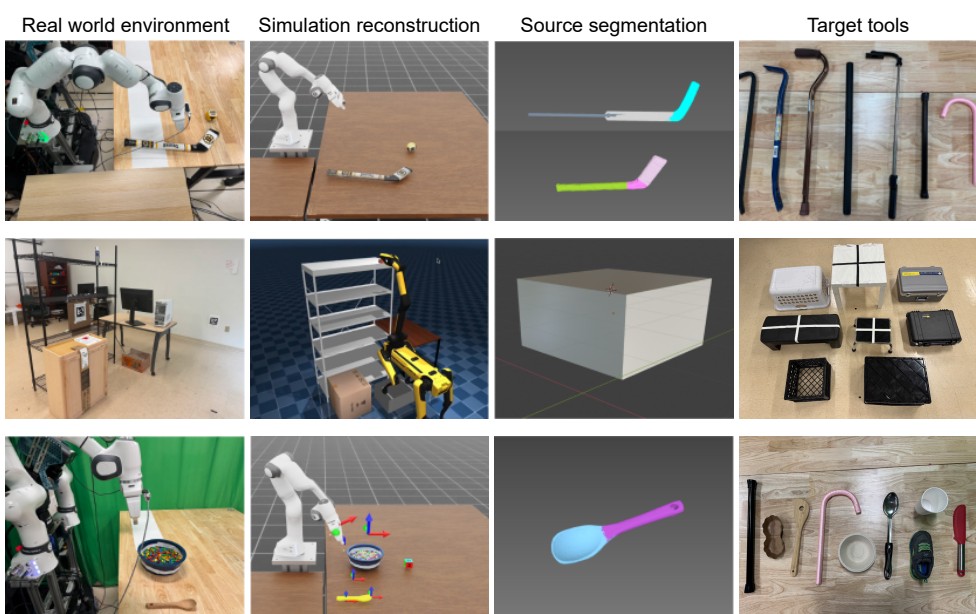

Figure 3: Top—Pulling the ball with a hockey stick. Middle—Reaching an object on the shelf using a platform. Bottom—Scooping candies in the bowl.

scores for keypoint transfer to different objects. (4) **Groundtruth Object (GT Obj.)** is an ablation baseline where all true target object meshes are reconstructed using (Code, 2025; Torne et al., 2024), and only the classification step is applied to them, i.e., after grasp and contact points detection the objects are imported to the simulator to check their success. (5) **Human**-annotated baseline: We collected responses from 22 participants, each completing a questionnaire under the same conditions as the feature suggestor. Specifically, they were presented with the task image, task description, and controller description, along with 12 VLM-generated candidate features, and asked to pick the causally relevant features to the best of their knowledge. [2]

### 4.1 EXPERIMENT DOMAINS

We compare our approach with baselines in three experiments that challenge the robot to reason about a diverse types of object and object-tool interaction. Note that the objects all have different primary functionalities in daily tasks.

**Table-top Pulling** requires the Franka Emika Panda robot to grasp a tool to pull an object located outside of its workspace (Figure 3, top). A suitable tool must meet both the task requirements like length and angled tip and also satisfy the robot's physical limitations, such as the payload. We assume that the robot has the skill to pull the toy puck with the toy hockey stick, and when it is in a new scenario where it needs to pick a new tool among the blue crowbar, selfie stick, walking cane, shepherd cane, black crowbar, yoga stick, curtain hanger. The task is satisfied if the puck is in reaching distance. Here, we run two versions of the task where we use the toy hockey stick model after reconstruction and just download and use a real hockey stick model as source objects.

The controller is written in terms of two keypoints: handle point and tip point. The robot grasps the tool at handle point, and brings the tip point behind the object and pulls the tool toward the initial grasp. Mass of the tool is determined by torque values of the robot after lifting it. The execution is stopped if the mass is over the working range found using the simulator.

---

[2]The data, prompts, code and human survey are on `https://github.com/toolanalogies/Toolanalogies`

**Table-top Scooping** requires the Franka Emika Panda robot to grasp a tool to dip it inside a bowl and scoop to obtain small candies (Figure 3, bottom). A suitable tool must meet both the task requirements like handle thickness and head curvature and also satisfy the robot's physical limitations, such as the payload. We assume that the robot has the skill to scoop some candies with a wooden spoon, and when it is in a new scenario where it needs to pick a new tool among black curtain rod, cartboard egg bite tray, wooden spatula, pink shepherd cane, cartboard bowl, metal serving spoon, plastic cup, toy shoe, and red scraper. The task is satisfied if the robot lifts any candy with the help of the tool.

The controller is implemented using 2 keypoints, handle and contact point. The robot grasps the object by the handle point, carries the tool over the bowl, then rotates the tool such that the tip point faces downwards, dips the tool into the bowl until the tip point is in contact with the candy, and lifts the tool back up while rotating it back to horizontal.

**Quadruped Reaching** challenges a Spot robot with an arm to retrieve an object from a shelf that is beyond its reach (Figure 3, middle). Unlike the pulling task, this requires the robot to use an object as a stepping tool to increase its reach and retrieve the object. A suitable tool must both be able to extend the robot's reach and also to fit between 2 obstacles to be placed. We used MuJoCo (Todorov et al., 2012) as the simulator. The candidate tools shown on middle in Figure 3 are a laundry basket, a milk crate, a stepping stool, a gripper box, and an aerobic step. The task is satisfied if the robot can reach to the object on the top shelf.

*The controller in the simulator*: We do not have access to the walking behavior that Spot has in the real-world, and as such, we spawned it on the platform in the simulator to resemble the behavior[3]. We control the arm of Spot using its kinematics model. Following Torne et al. (2024), we downloaded similar items for the shelf and the goal object from the internet.

*The controller in the real world*: We utilize the Boston Dynamics Spot SDK to control the robot. The skill is written as moving forward by a specified distance towards the goal. During this forward trajectory, Spot autonomously detects and steps onto an obstacle, in this case, a stepping tool, without requiring explicit modification by the built-in controller, which again is not exposed to outside. Following this movement, we command Spot's onboard arm to reach to the shelf by pose commands. Note that there is no keypoint for the stepping behavior, so keypoint baselines are removed.

## 4.2 TASK PERFORMANCE AND INFERENCE TIME ANALYSIS

Table 1: Task performance across methods

|  | Ours | o3 | o3 (RGB-D) | o3 (t-PCL) | DINOv2 | MAGIC | CoTDet | GT Obj. |
|---|---|---|---|---|---|---|---|---|
| Table-top Pulling | 100% | 50% | 0% | 50% | 50% | 50% | 50% | 100% |
| Table-top Scooping | 50% | 0% | 0% | 0% | 0% | 0% | 0% | 50% |
| Quadruped Reaching | 66.7% | 0% | 33.3% | 33.3% | N/A | N/A | 66.7% | 100% |

Table 1 shows the success of the methods in predicting suitable tools among all objects in the environment. Ground-truth values of all target tools were obtained by executing the adapted controller in the real robot setup. Benchmarks with preference lists were evaluated by treating the top-2 tools (out of 8) as successes and the rest as failures. Toolanalogies achieved higher accuracy than all baselines except the ablation with ground-truth object models.

Across experiments, vision-only embeddings (e.g., Dino features, VLM image prompts) captured coarse shape cues but failed to account for physical task constraints, often ranking payload-inappropriate tools (e.g., crowbars) too highly. Adding local curvature descriptors (Magic) improved sensitivity to contact geometry but overemphasized curvature at the expense of other causal cues (length, flatness), reducing overall performance. Point-cloud context allowed VLMs to better reason about length and thickness, but gaps in physical reasoning (weight) persisted. Dataset-driven detectors (e.g., CoTDet) excelled when familiar objects appeared, but generalization to novel yet

---

[3]For the real-world walking behavior, we contacted RAI (Boston Dynamics, 2024) but could not obtain the controller in simulation.

Table 2: Human supervision cost, and interpretability

|  | Ours | o3 | o3 (RGB-D) | o3 (t-PCL) | MAGIC | CoTDet | GT Obj. |
|---|---|---|---|---|---|---|---|
| Human supervision | ✗ | ✗ | ✗ | ✗ | ✗ | ✗ | ✓ |
| Interpretability | ✓ | ✓ | ✓ | ✓ | ✗ | ✗ | ✗ |

functionally valid tools (e.g., laundry basket) remained weak. In the final experiment, where objects exhibited very different and unknown dynamics such as deformability, performance of the ground-truth object baseline degraded, whereas our method still identified a viable option through causal reasoning.

In sum, baselines reveal recurring gaps: (i) missing physical priors (mass), (ii) reliance on single visual heuristics (curvature or semantics), and (iii) limited transfer from richer geometry without task-level reasoning. Our method addresses these by analyzing causal shape features via 3D object editing and testing physical features in a simulator.

We assume only the source object model, reasoning about target objects through semantic features. This allows to interpret the reasoning behind the tool selection, which is achieved only by VLM baselines. However their reasoning is physically ungrounded and untested unlike ours. In addition, although baseline object-truth modeling outperforms other methods, it requires minutes to scan a single object. Moreover, 3D modeling demands unoccluded multi-view images—often needing operator assistance—whereas our method avoids this. Table 2 illustrates these differences. Failure cases of our method are detailed in 4.5.

### 4.3 CAUSAL FEATURE ANALYSIS AND HUMAN ALIGNMENT

Table 3 shows the contribution of the features to the task success among all inferred causal features, and the similarity of inferred causal features to the causal features suggested by the human study. We labeled a feature 'causal' for human baseline if more than 50% of participants agree. We report the percentage of overlapping features with our pipeline. See A.7.

Table 3: Causality analysis

|  | Human-alignment | Contributing features to real-world success |
|---|---|---|
| Pulling | 83% | 3/5 |
| Reaching | 83% | 4/5 |
| Scooping | 57% | 2/4 |

We see that for all tasks, our approach finds the causally related features to classify tools; however, this does not mean that it discovers all the causal features detected by humans. During the survey, humans labeled synonym features as causal, but our pipeline investigates features incrementally after the initial 6, so it is less likely to discover synonym features. Since target objects are not known during feature generation, it is possible for both humans and our method to label unnecessary features for the task. Since humans see more features during the survey, they are more likely to mention features that will not be discovered by our pipeline. It is also possible for humans to mention features not supported by the current feature editors, such as friction.

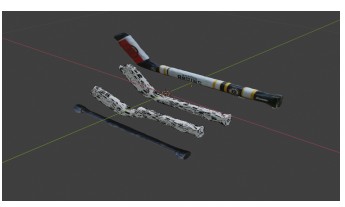
(a) Tool matching success

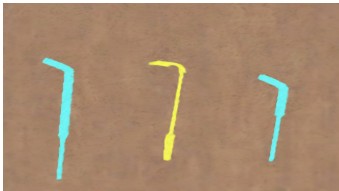
(c) VLM classification

(b) Tool matching failure

Figure 4: Classification methods

## 4.4 Classification Metrics

In the Figure 4, we explain the different classification methods mentioned throughout this paper. Image a illustrates our classification approach. We transform our source object (hockey stick) through two successive operations designed to approximate the geometry of a curtain rod. In contrast, Image b presents a failure case of the proposed classification approach. This failure arises from a mesh exhibiting disproportionately distributed mass and surface area, which misleads the classification algorithm. Image c illustrates the alternative approach for such failures. We render the partial point-clouds of the target tool and the two boundary source tools that bracket the continuous operational range for the feature under consideration. Here, the pointclouds of the shortest and longest suitable sticks are shown in blue. Any stick whose length falls between them is also suitable.

## 4.5 Failure Cases

**Feature suggestor failure:** Vision Language Model (VLM) may not list all possible causal features at first, therefore we included more iterations into our pipeline. For example, our feature suggester overlooked tool mass at first iteration as other VLM or vision approaches. However, the pipeline asked for additional features and fixed the mistake of VLM. If VLM cannot even produce the causal feature in more iterations, that feature cannot be learned.

**Object editor failure:** Although we do not define any variables to do causal inference, and we get them from a VLM (which is an important contribution in our opinion), our editing ability is limited by the capability of the ParSEL framework and the dynamics models. To mitigate this, we condition the prompts on the editor's grammar to encourage compatible outputs. If failure persists, it cannot be addressed, we skip to the next target object to solve the task. The framework is totally compatible with replacing the dynamics model or the object editor function later if better models are available to improve on this limitation.

**Failures due to the controller and sim2real gap:** It is possible that simulation dynamics is not exactly the same as in the real world. In the literature, simulation tuning with real-world data is used to solve this issue. However, it is out of the scope of this project. In such scenarios, we skip to the next target object to solve the task. Similarly, the controller may not adapt to every novel object although we are employing a state-of-the-art method. Comprehensive adaptation methods (e.g., reinforcement learning) are promising but beyond our current focus. We want our pipeline to run one-shot, and we skip to the next target object to solve the task for such failures.

**Segmentation failure:** The scene segmentation or the part segmentation may fail. In our work, we are assuming that they are good enough for our experiments and better models can be plugged in for more complex environments.

**Pointcloud classification limitation:** Since we are using a single view pointcloud of the target object to do classification using Chamfer distance, optimizing for thickness features become a limitation. If the task depends on thickness, e.g. for grasping, all objects will result in failure. We used heuristics like producing a syntetic pointcloud by taking symmetry of the pointcloud along xy plane, and merging the two pointclouds.

## 5 Conclusion

We introduced a causal reasoning framework for creative tool use that identifies and repurposes novel objects as tool substitutes in diverse scenarios. During training, the robot performs vision-based mesh reconstruction to generate a physical reconstruction of the scene in a physics-based simulator. The simulator serves as a causal reasoning engine, where we systematically intervene on semantic properties of tools and evaluate the effect on task success to identify the causally relevant features. These features are then used by a classifier, which selects the most suitable substitute tool from the available objects. Our approach generalizes to novel objects by leveraging the commonsense reasoning of VLMs, is grounded in physics by our use of physics-based simulation and provides interpretable justification by virtue of causal reasoning. Real-world experiments in three table-top and mobile manipulation domains show that it outperforms baselines in creative tool use, significantly enhancing the open world manipulation capabilities of robots.

## ETHICS STATEMENT

We collected human responses on our experiments to understand the alignment between the causal features assessed by humans and our system. The responses to the questions in the survey consist of a number of multiple choice selections, and as such, there are no identifying information in the collected data. We made sure to remove possibly identifying information in the feedback form of the survey when there are any. Nevertheless, we keep this feedback private. In making this survey, we adhered to the ICLR Code of Ethics, rules enforced by the countries and affiliations of authors. Our affiliated institutions' self-determination tool clearly implies that this survey does not need an IRB review as we do not collect any identifying information.

## REPRODUCIBILITY STATEMENT

As our method consist of several components, the reproducability of the method becomes even more important for assessing the significance of the results. All of our data, prompts, procedures, and human survey are available at `https://github.com/toolanalogies/Toolanalogies`.

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

# A APPENDIX

## A.1 A MOTIVATING EXAMPLE

Consider the toy grid environment in Figure 5 with objects listed on the right. The task is to move the green ball to the goal while avoiding the lava tile. When the green ball is on lava, the agent must first free the ball by pushing the brown box—an intermediary object—onto the lava tile, and then move it to the goal. In a new environment in which there are no boxes, the agent must find another object that can functionally substitute for the box. Trying each object as the replacement might take an arbitrarily long time in environments with many objects, especially when the evaluation is costly (e.g., the use of a dynamics model) and poses a safety concern (e.g., stepping on the lava tile). It would be reasonable to pick objects based on their features if only the agent knew which of those are causally relevant. However, the agent must conduct interventional experiments (Pearl, 2009) to find out such features. For instance, by changing the features of the box—by an intervention—and observing its effects, the agent can identify that the replacement for the box should be rigid and movable while its color is irrelevant, and figure out that the red ball can be used instead. While changing the object properties in game-like environments is possible, doing so in real world scenarios is non-trivial. In this paper, we demonstrate that a similar approach can address real-world tool manipulation problems in which the features cannot be listed as in this example and cannot be easily manipulated in the real world.

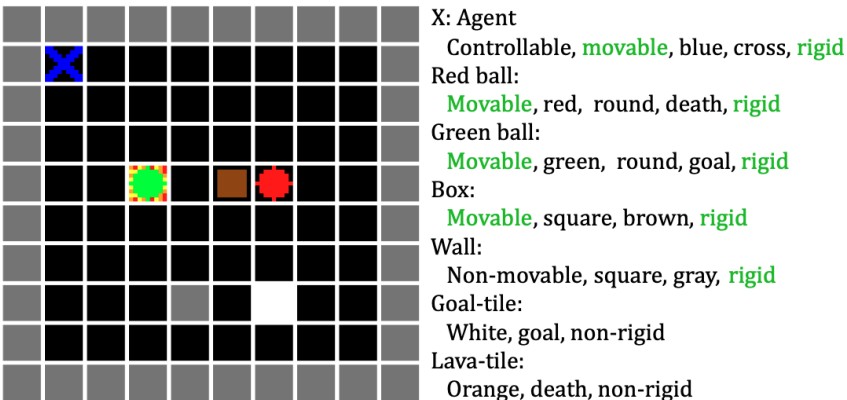

Figure 5: Toy grid-world used to illustrate our problem setting in an idealized scenario where object features are pre-listed and directly editable unlike real-world: the blue X agent navigates walls and interacts with objects whose attributes (movable, goal, death, etc.) are listed on the right.

## A.2 EXTENDED RELATED WORK

Tool use has traditionally been studied under the topic of affordances (Gibson, 1979). In push and pull tasks with various shaped sticks, affordances are found by clustering object-effect categories using predefined actions or motor babbling (Sinapov & Stoytchev, 2008; Stoytchev, 2005; Tikhanoff et al., 2013). To achieve tool transfer beyond categorization, object features are learned with virtual tools in simulation (Mar et al., 2017; Nishide et al., 2012; Takahashi et al., 2017; Tekden et al., 2024). Gonçalves et al. (2014) investigated the role of predefined shape descriptors with visual feature learning. These methods haven't been scalable to different tools due to the limitation of collecting robot experience at the order of the complexity of tool use tasks.

Keypoint representation is another widely used approach to select suitable tools and manipulation policies (Manuelli et al., 2022). Tee et al. (2022) transfers keypoints on robot limbs to tools to attribute limb functionality. Mao et al. (2023) used contact points on objects and task and motion planning (Garrett et al., 2021) in balancing tasks in addition to push and pull tasks. Gao & Tedrake (2021) coupled key points with a feedback controller for wiping and peg insertion. These works addressed tool use problems where tools look similar and arguably belong to the same category.

To accomplish transfer between intercategorical tools, part-level affordances are sought (Myers et al., 2015; Schoeler & Wörgötter, 2016; Kroemer et al., 2012). Fitzgerald et al. (2019; 2021)

used human correction trajectories to transfer tooltip's pose constraints for hooking, sweeping, and hammering tasks. In Agostini et al. (2015), an affordance knowledge base is used for tool substitution to satisfy plans in salad-making tasks. We are using VLMs similarly as a vast knowledge base. However, we are checking and grounding suggestions by reconstructing the task in simulation.

End-to-end methods were also used to predict actions using visual features to use different tools in sweeping and hammering tasks (Fang et al., 2018; Xie et al., 2019). In addition, task-specific key point predictors, along with procedural tool generation, have shown improvements in intercategory tool use in pushing, reaching, and hammering tasks. Task information was provided as environment keypoints (Qin et al., 2020) or as a reward function to the network (Turpin et al., 2021). In Qin et al. (2020); Turpin et al. (2021); Fang et al. (2018), tools of the X, L and T shape were generated procedurally by combining convex parts. Instead, we use semantic information from LLMs for tool generation so that generated tools are directly related to semantic features, and we do not need any additional training to learn affordance features. In addition, these procedural generation methods (Qin et al., 2020; Turpin et al., 2021; Fang et al., 2018) used 600, 10000, and 18000 tools, respectively, for training compared to our 120. In Liu et al. (2024); Qin et al. (2021), global and local geometric features are used together to transfer contact points and for tool selection. Although this approach allows transfer between tools that have geometrically similar parts (it requires demonstrations, and) it does not reason about physical features and environment constraints.

In Abelha & Guerin (2017); Gajewski et al. (2019), pointclouds are processed to characterize tools, grasp and contact segments of the tool are found flexibly based on the task. Abelha & Guerin (2017) used CAD-models from web to populate a dataset of 5000 tools to test in simulation, and a Gaussian Process is fitted to their affordance score, to classify new objects later for tool selection. This approach was tested in cutting, lifting, hammering, and rolling tasks. Using web models is an alternative to tool generation, but again, our approach does need additional training with thousands of tools to find affordances due to semantically meaningful generation. Zhu et al. (2015) considers predefined physical features for the task differently from previous work, which we show is essential for robot tool use.

Recently, Visual Language Models have been used to benefit from web-scale data for tool affordance. The line of work Tang et al. (2023); Huang et al. (2024); Yu et al. (2024) in the vision community uses the similarity between the text encoding of general knowledge of tools queried from VLMs, and the visual encoding of task images to propose and segment a suitable tool for the task. Although we also consider VLMs to be great resources for general information on prior experiences, using them alone is problematic in robotics. By definition, they lack interaction data for the current task, therefore knowledge of task dynamics, causal relationships, and the action capabilities of the agent.

In robotics, VLMs have gained attention in tool selection and manipulation policy as well. Ren et al. (2023) uses language descriptions of tools as affordance features to train a meta-policies for sweeping, hammering, pushing, and lifting tasks. In Lee et al. (2024), Large Language Models(LLMs) are used as a high-level symbolic planner for bimanual pull and push tasks. In Car et al. (2024), high-level planner was combined with a low-level planner and vision modules to use tools with LLMs, but tool selection is not addressed, and affordance prediction was limited with grasp point prediction. Robotool (Xu et al., 2024), closest to our work, used LLMs to analyze the problem to extract key concepts, create plans selects a tool, and execute parametrized skills for reaching, grasping, and pressing tasks. The framework uses privileged task information such as the layout of objects, the positions, sizes, and physical properties of objects, grasp points on objects, as well as robot and environment constraints. On the other hand, we get the accurate reconstruction of only the source tool, physical properties are either inferred through interaction or not used, and satisfying robot and task constraints is learned through a simulator without any explicit definition. Other methods like Lin et al. (2025); Gao et al. (2025) address tool design problem (Liu et al., 2023) via VLMs. Here, we are trying to repurpose an everyday object to satisfy the task instead of designing and printing 3D shapes.

## A.3 COMPUTER SPECIFICATIONS

The computer that we used to run our pipeline has Ubuntu 20.04 as the operating system, Nvidia A6000 as the GPU, AMD Ryzen threadripper 7970x, as the CPU, and 128 GB for the working memory.

## A.4 AUTOMATIC PART SEGMENTATION

To split the object into semantically meaningful parts, we explored existing research focused on part segmentation without requiring manual annotations, in order to preserve the autonomous nature of the pipeline.

We identified SAMPART-3D as a pioneering method well-suited to our goals. SAMPART-3D segments 3D objects into multi-granularity parts without any part-level annotations.

SAMPART-3D outputs split point cloud meshes at varying levels of granularity, ranging from 0.0 to 2.0. We found that a granularity scale of 1.5 worked best for our use case. The segmented object parts are labeled using different colors. To separate these parts, we grouped the point cloud features by color (each color representing a distinct object part) and converted each grouped point cloud into a triangulated mesh.

Figure 9 illustrates the outputs of SAMPART-3D. The left image is part segmentation of the source toy hockey stick. It segmented stick tip and stick body separately, such that the object editor can apply edits for features like tip width, tip angle. The middle image shows that part segmentation fails for the hockey stick downloaded from the Web. It does not allow similar edits because the tip is segmented together with the body. This is a limitation of this model, it uses SAM as the backbone. Therefore, it cannot detect parts if there is not enough color or texture change. The right image shows the result after the manual fix which is to paint the tip of the stick.

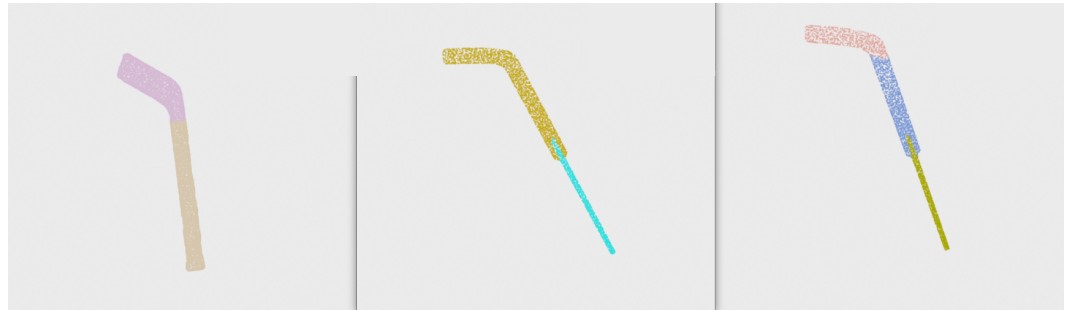

Figure 6: SAMPART-3D Output

## A.5 OBJECT EDIT BY SEMANTIC FEATURES

**ParSel Ganeshan et al. (2024)** is a system that enables users to precisely edit 3D assets using natural language prompts. It takes a segmented 3D mesh and an edit request to generate a parameterized editing program, allowing to change the mesh with a controlled magnitude. While LLMs identify the initial editing operations, ParSEL computes Analytical Edit Propagation (AEP) algorithm which integrates computer algebra for geometric analysis to propagate the initial edit to the rest of the object coherently. This approach effectively creates stable shapes without defects that are successfully imported into the dynamics model later.

The data created from the Automatic Part Segmentation is used as the mesh inputs. To annotate the parts, we used a Visual Large Language Model (VLM) to label parts segmented by color (the output from SAMPART-3D was used as input to the VLM).

We then store the outputs of the part edits at different scales of edit to produce a collection of tools with varying edits of the same characteristic. This process is performed repeatedly for all characteristics that must be checked.

### A.6 Classification by Chamfer Distance and optimization

To match a mesh to the reference mesh, we use mesh editing to maximally match the reference respective point clouds of the two meshes. The metric we use to measure the similarity is Chamfer distance.

$$\text{chamfer}(P_1, P_2) = \frac{1}{2n} \sum_{i=1}^{n} \|x_i - \text{NN}(x_i, P_2)\|$$

$$+ \frac{1}{2m} \sum_{j=1}^{m} \|x_j - \text{NN}(x_j, P_1)\|$$

We first perform one operation or edit with Parsel. (Parsel provides a range of values with varying degrees of editing). We then sample this scale with a fixed granularity at uniform distances. These meshes are then sampled to create point clouds, which are then compared to the reference mesh's point cloud. The point cloud with the lowest chamfer distance is then saved, and the consequent Parsel operation is performed on this point cloud. This process is repeated until you get a point cloud that matches the reference point cloud well enough. Note that operations are run sequentially.

### A.7 Human survey results

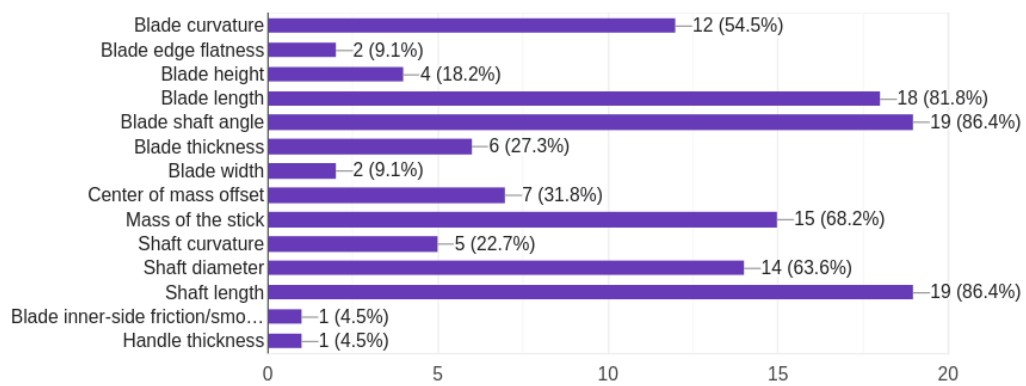

Figure 7: Human survey results for pulling task

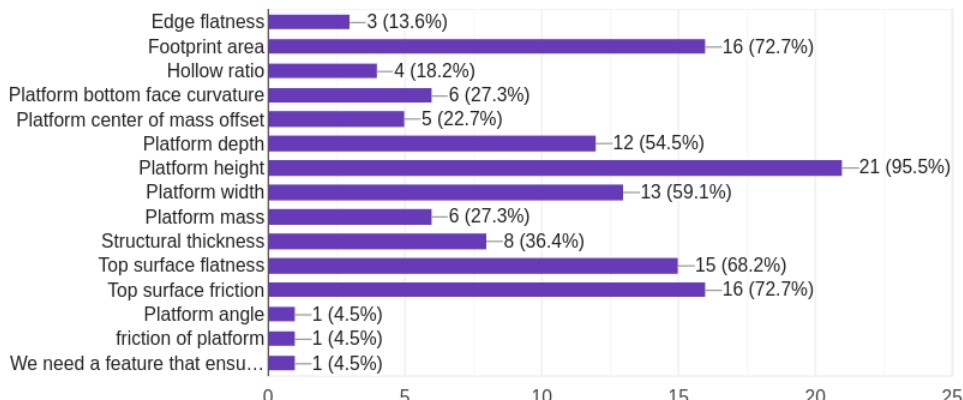

Figure 8: Human survey results for reaching task

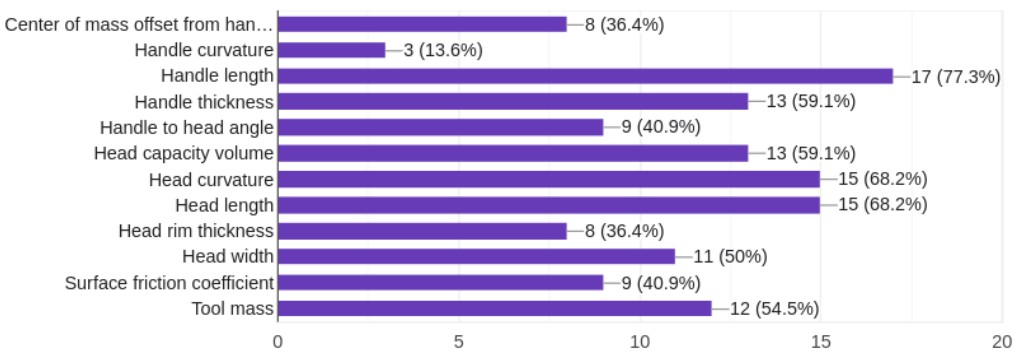

Figure 9: Human survey results for scooping task

## A.8 PROMPTS

### A.8.1 FEATURE GENERATION PROMPTS

Developer Prompt

```
Your goal is to help robots classify tools in order to solve tasks. There
    will be two outputs from this process. A list of prompts for a shape
    editor to modify a prototypical object, and a list of generic
    features that will be used to identify tool suitability. You will be
    provided with the image of the task in which the source tool is
    present, an explanation of the robot skill executed with the source
    tool, and examples of shape editing prompts with guidelines. To help
    the robot, observe that it successfully completes the task using the
    source tool in the image, and list numerical properties of that tool
```

that are causally relevant to task success, i.e., properties whose
change would help or hinder the tool's ability to contribute toward
the objective. There are two main kinds of simulatable properties we
are interested in: physical features and shape features.
Physical features include mass, friction, moments of inertia, etc. For
now, the only physical property you can use is mass, since that's all
the simulator supports right now.
The second class of properties is shape features. Shape features should
be generic, numeric, and real-valued. A good shape feature does not
reference overly specific parts which only some of the objects have,
but instead refers to more generic attributes. In order to generate
the prompts for editing the prototypical tool, these generic features
should be refined into simulatable modifications, and then they'll
be passed into a shape editor which will allow the robot to imagine
modifications to a prototypical tool, and find the feasible range of
values for each feature. The features should be general enough to
apply to all objects equally, but when writing the prompts for shape
editing, reference only the prototype object, down to the detail of
axis for rotation, etc. We will later use these features to select
the appropriate substitution tool for the task. Therefore, try to
only list features that can be estimated for each tool, and which
numerical variation will help or hinder the robot at achieving the
task.
Attached is a json containing the part-segmentation for the prototypical
tool. Be sure to express the shape edit request in terms of relevant
parts of the prototypical object actually contained in the input json
.
In order to select features, think about, and then list twelve different
shape features. Do not mention synonyms as different features. Rank
them in the order in which they will 'most likely to make or break
the task'. Once you've listed and ranked the tools and features, put
the top six features into a json along with 12 candidate features as
in the example below.
Finally, once you have listed the features, you will be creating shape
edit requests for each shape feature. Shape request order should
match the order of the final features in the json. Your request will
be sent to another LLM which will be writing the shape-edit program.
Keep this in mind, and try not to reference parts, features of the
object which are not available to the simulator, or other non-
standard terminology in your prompt. Be sure to explain your
reasoning concisely to the other LLM, so it gets the programming
right.
It is very important that you don't make json syntax error. This is a
example of the JSON file you should return:

List of features by 'make or break' criterion:
```json
{
"candidate_properties": [
  {
  "name": "feature1_name",
  },
  {
  "name": "feature2_name",
  },
  ...
  {
  "name": "feature12_name",
  },
],
"final_properties": [
  {
  "name": "feature2_name",
  },
  {
```

```
"name": "feature5_name",
},
...
],
"shape_prompts":[
  {
  "part":part_name,
  "edit_request":request2_text",
  },
  ...
]
}
```
I picked feature2_name because... I picked feature5_name because...

"""

User Prompt

user_prompt_cube_retrieval = """Your task is to help a Franka Emika Panda
    robot arm retrieve a hockey ball. Attached is an image of the task
   with a hockey stick (source tool).
 The controller is implemented using 2 keypoints, handle and contact
    point. The robot grasps the object by the handle point, carries the
    contact point behind the ball,
 then pulls the tool back to bring the ball closer. Here is the json
    describing the parts of the prototypical tool, available for editing
    :
 ```json
 [
   {
     "objs": [
      "hockeystick_blade",
      "hockeystick_shaft"
     ],
     "name": "object",
     "text": "object",
     "children": [
       {
        "objs": [
          "hockeystick_blade"
        ],
        "name": "hockeystick_blade",
        "text": "hockeystick_blade",
        "children": []
       },
       {
        "objs": [
          "hockeystick_shaft"
        ],
        "name": "hockeystick_shaft",
        "text": "hockeystick_shaft",
        "children": []
       }
     ]
   }
 ]
 ```

 """

  user_prompt_shelf_reach = """Your task is to help a Boston Dynamics Spot
       robot reach for a blue bottle on a partially obstructed high shelf.
```

```
We will attempt to place the selected object between some obstacles at
    the base of the shelf. Attached is an image of the task, the
    controller is implemented as walking forward to the shelf, stepping
    on the tool, and then reaching up with the robot arm to grab the
    bottle.
Here is the json describing the parts of the prototypical tool,
    available for editing:
```json
[{
  "objs": ["cube_triangulated", "cube_dummy"],
  "name": "cube_master",
  "text": "cube_master",
  "children": [
    {
      "objs": ["cube_triangulated"],
      "name": "cube_part",
      "text": "cube",
      "children": []
    },
    {
      "objs": ["cube_dummy"],
      "name": "dummy",
      "text": "dummy",
      "children": []
    }
  ]
}]
```
"""

user_prompt_scooping = """Your task is to help a Franka Emika Panda
    robot arm get some candy from a bowl. Attached is an image of the
    task with a wooden spoon (source object),
The controller is implemented using 2 keypoints, handle and contact
    point. The robot grasps the object by the handle point, carries the
    tool over the bowl,
then rotates the tool such that the tip point faces downwards, dips the
    tool into the bowl until the tip point is in contact with the candy,
    and lifts the tool back up while rotating it back to horizontal.
Here is the json describing the parts of the wooden spoon, available for
    editing:
```json
[
  {
    "objs": [
      "spoon_head"
      "spoon_handle"
    ],
    "name": "object",
    "text": "object",
    "children": [
      {
        "objs": [
          "spoon_head"
        ],
        "name": "spoon_head",
        "text": "spoon_head",
        "children": []
      },
      {
        "objs": [
          "spoon_handle"
        ],
        "name": "spoon_handle",
```

```
1188            "text": "spoon_handle",
1189            "children": []
1190          }
1191        ]
1192      }
1193    ]
       ```
1194    """
```

## A.8.2 CLASSIFICATION PROMPTS

```
I am gonna send you pictures, and a feature name.
In the picture, blue objects show lowest value and highest value that the
    feature can get,
and the yellow object is the candidate object.
You need to tell me in one word (True/False)
if yellow objects's feature satisfies the range defined by blue objects.
```

## A.8.3 BASELINE PROMPTS

Baseline with RGB only

```
tool_selection_cube_retrieval = """Your task is to help a Franka Emika
    Panda robot arm retrieve a hockey ball. Attached is an image of the
    task (which also contains a wooden spoon as the prototypical object)
    ,
as well as an image containing various suitable real-world tools. Here
    are the names of the tools that should be in the image:
black iron crowbar, blue iron crowbar, walking cane, yoga stick, selfie
    stick, curtain rod, shepherd cane.
The controller is implemented using 2 keypoints, handle and contact
    point. The robot grasps the object by the handle point, carries the
    contact point behind the ball, then pulls the tool back to bring the
    ball closer.
In the previous conversation, you have been instructed to provide a list
    of causally relevant features for the task.
The answer is sampled 10 times as we are giving you the most repeated
    answer. Now, you need to find a substitution tool. You are given
    with the task image with the source tool, the tool image with
    various real-world tools, and the list of the found causal features.
First, list all the tools in the image. Only include tools that are
    actually in the image in the list. Consider the found causally
    relevant features and rank the tools in the order in which they will
    'most likely' be suitable for solving the task. Tell if the tool is
    suitable or not.
Report the rank and success booleans as json file and provide a short
    explanation like this, it is very important that you don't make json
    syntax error:
```json
{
"tool_ranking": [
    {
        "name": "tool1_name",
        "success": true,
    },
    {
        "name": "tool2_name",
        "success": true,
    },
    ...
    {
        "name": "tooln_name",
        "success": false
    }
```

```
            ]
        }
        ```
    I picked tool1_name because... I picked tool2_name because...
    """

    tool_selection_shelf_reach = """Your task is to help a Boston Dynamics
        Spot robot reach for an orange cube on a partially obstructed high
        shelf.
    We will attempt to place the selected object between some obstacles at
        the base of the shelf. Attached is an image of the task, as well as
        an image containing various suitable real-world tools.
    The controller is implemented as walking forward to the shelf, stepping
        on the tool, and then reaching up with the robot arm to grab the
        cube.
    Here are the tools that should be in the image: white IKEA coffee table,
         step stool, square milk crate, rectangular milk crate, aerobic
        stepper, grey gripper case, laundry basket, black gripper case, and
        GPU packaging box.
    In the previous conversation, you have been instructed to provide a list
         of causally relevant features for the task. The answer is sampled
        10 times as we are giving you the most repeated answer.
    Now, you need to find a substitution tool. You are given with the task
        image with the source tool, the tool image with various real-world
        tools, and the list of the found causal features.
    First, list all the tools in the image. Only include tools that are
        actually in the image in the list. Consider the found causally
        relevant features and rank the tools in the order in which they will
        'most likely' be suitable for solving the task. Tell if the tool is
        suitable or not.
    Report the rank and success booleans as json file and provide a short
        explanation like this, it is very important that you don't make json
        syntax error:
    ```json
    {
    "tool_ranking": [
        {
            "name": "tool1_name",
            "success": true,
        },
        {
            "name": "tool2_name",
            "success": true,
        },
        ...
        {
            "name": "tooln_name",
            "success": false
        }
        ]
    }
    ```
    I picked tool1_name because... I picked tool2_name because...

    """

    tool_selection_scooping = """Your task is to help a Franka Emika Panda
        robot arm get some candy from a bowl. Attached is an image of the
        task (which also contains a wooden spoon as the prototypical object)
        ,
    as well as an image containing various suitable real-world tools. Here
        are the tools that should be in the image:
```

```
black curtain rod, cartboard egg bite tray, wooden spatula, pink
    shepherd cane, cartboard bowl, metal serving spoon, plastic cup, toy
     shoe, red scraper.
The controller is implemented using 2 keypoints, handle and contact
    point. The robot grasps the object by the handle point, carries the
    tool over the bowl,
then rotates the tool such that the tip point faces downwards, dips the
    tool into the bowl until the tip point is in contact with the candy,
    and lifts the tool back up while rotating it back to horizontal.
In the previous conversation, you have been instructed to provide a list
    of causally relevant features for the task.
The answer is sampled 10 times as we are giving you the most repeated
    answer. Now, you need to find a substitution tool. You are given
    with the task image with the source tool, the tool image with
    various real-world tools, and the list of the found causal features.
First, list all the tools in the image. Only include tools that are
    actually in the image in the list. Consider the found causally
    relevant features and rank the tools in the order in which they will
    'most likely' be suitable for solving the task. Tell if the tool is
    suitable or not.
Report the rank and success booleans as json file and provide a short
    explanation like this, it is very important that you don't make json
    syntax error:
'''json
{
"tool_ranking": [
    {
       "name": "tool1_name",
       "success": true,
    },
    {
       "name": "tool2_name",
       "success": true,
    },
    ...
    {
       "name": "tooln_name",
       "success": false
    }
    ]
}
'''
I picked tool1_name because... I picked tool2_name because...

"""
```

Baseline with RGB and Depth Images

```
Insn the image, you can see a task for the Franka Emika robot:
it needs to pick up the ball, but the ball is out of reach.
You must determine which tool can be used to pull the ball toward the
    robot.

The .pkl file structure is as follows:
snapshot = {
    "stamp":       stamp.to_sec(),
    "rgb":         rgb,       # uint8 BGR
    "depth":       depth,     # float32 metres
    "K":           K,         # 3x3 intrinsics
    "T_cam2world": T,         # 4x4 float64
    "frames": {
       "rgb": rgb_msg.header.frame_id,
       "depth": depth_msg.header.frame_id,
       "world": self.world_frame
    }
```

```
}
The scene.pkl file describes the scene.

Other .pkl files has the same format for the scene where a replacement
    candidate object
(one of black iron crowbar, blue iron crowbar, walking cane, yoga stick,
    selfie stick, curtain rod, shepherd cane)
is placed next to the hockey stick (the known tool that works for the
    task) on the table.

I also add a .pkl just for hocket stick on the table, in case it will be
    useful.

Can you read the .pkl files, analyze the data, and rank the candidate
    sticks
from 1 to 7 according to their suitability for completing the task?
```

### Baseline with Pointclouds

```
In the image, you can see a task for the Franka Emika robot:
it needs to pick up the ball, but the ball is out of reach.
You must determine which tool can be used to pull the ball toward the
    robot.

The .pkl file structure is as follows:
snapshot = {
    "stamp":      stamp.to_sec(),
    "rgb":        rgb,      # uint8 BGR
    "depth":      depth,    # float32 metres
    "K":          K,        # 3x3 intrinsics
    "T_cam2world": T,       # 4x4 float64
    "frames": {
        "rgb": rgb_msg.header.frame_id,
        "depth": depth_msg.header.frame_id,
        "world": self.world_frame
    }
}
The scene.pkl file describes the scene.

The .pcd files are pointclouds in the frame of robot base:
one for the hockey stick (the known tool that works for the task)
and one each for the black crowbar, blue crowbar, walking cane, yoga
    stick,
selfie stick, curtain rod, and shepherd cane as candidate replacement
    tools.

Can you read the .pkl and .pcd files, analyze the data, and rank the
    candidate sticks
from 1 to 7 according to their suitability for completing the task?
```