# OpenReview forum: "Creative Robot Tool Use by Counterfactual Reasoning"
_ICLR.cc/2026/Conference — Submitted to ICLR 2026_

### Official Review · Reviewer_z4ew · 2025-10-16

**Soundness:** 2
**Presentation:** 2
**Contribution:** 2
**Rating:** 2
**Confidence:** 3

**Summary:**

This paper proposed a causal reasoning framework that helps robots to find appropriate tools from unseen candidates for a specific task. During training, the authors first asks a VLM to propose features of an object that are relevant to completing the task, then new objects are synthesized by altering these features. These new objects are then used in simulation to figure out the actual importance of these features, which are used by classifiers to determine the right tool during inference.

**Strengths:**

1. The authors leverage VLM's commonsense knowledge to help robots complete tasks, which is a promising direction.
2. Real-world test setups were established.

**Weaknesses:**

1. The proposed approach did not appear sufficiently generalizable to more complicate tasks and objects. The experiments are conducted on only three tasks.
2. The illustrations are difficult to understand. Particularly on the right side of Fig.2, the title suggests "classification" but it is unclear where the classification step is at. And why is the same image of the novel object is considered "approximately equal to" to different shapes? Are the two shapes "approximately equal" too?
3. Typos. In Fig.2-left, the feature "suggestor" (which should be suggester?) is labelled with (e), whereas in the texts, it is labelled with $\phi$. In Line 222, I did not find grasp points in Fig.3, so I assume it should be Fig.2?

**Questions:**

1. The authors use simulations with modified 3D objects to determine whether the suggested features are relevant to the task. Is it possible to replace this procedure with VLMs, which would be less time-consuming.
2. In Line 194, what are the 12 features in "top 6 most voted features out of 12"? Are they fixed for all tasks?
3. With the relevant features, how does the classification step work? The texts in the paper are not very clearly written.
4. Sec 4.2 is titled "Task Performance and Inference Time Analysis", where is the inference time analysis? I only find one sentence with time-related information ("it requires minutes to scan a single object"), and it is not about inference.

---

> ### Author Response · Authors · 2025-11-18
> **Response to Reviewer z4ew**
>
> We thank Reviewer z4ew for carefully engaging with our work and for their time. We are glad they see leveraging VLM commonsense knowledge for robot tool use as a promising direction and appreciate the real world experimental setups included in the paper. We answer their concerns and questions below:
>
> >The proposed approach did not appear sufficiently generalizable to more complicated tasks and objects. The experiments are conducted on only three tasks.
>
> We appreciate the concern about generality, but we feel the current scope is broader than the review suggests.
>
> Our method is task‑agnostic by construction: the only task‑specific inputs are a natural language description of the task and controller and a single source tool demonstration. In terms of embodiment and task diversity, we already evaluate our pipeline on two different robots for three qualitatively different tool‑use scenarios: a Franka arm for table‑top manipulation (pulling, scooping) and a Boston Dynamics Spot with an arm (stepping).
>
> In all three, we use different source tools and different candidate object sets, and the method discovers different causal features per task. Therefore, we believe that our framework is generalizable and not tied to specific tools and tasks.
>
> We would be happy to show how our framework could extend or where it would currently fail, if you let us know what type of tasks and objects that you have in mind.
>
> >The illustrations are difficult to understand. Particularly on the right side of Fig.2, the title suggests "classification" but it is unclear where the classification step is at. And why is the same image of the novel object is considered "approximately equal to" to different shapes? Are the two shapes "approximately equal" too?
>
> Thank you for the question, we fixed our figure to clarify the classification step and obtaining the source object model step. We will also fix the text to clarify these points as below:
>
> On the right side of Fig. 2, the sequence is:
>
> - Causal features that survived the counterfactual tests (e.g., length, tip angle).
> - Novel object: RGB‑D of the candidate tool.
> - Object editor + Chamfer distance: we edit the source tool along each causal feature to make it geometrically as close as possible to the novel object (minimizing Chamfer distance in point‑cloud space).
> - The edited source tool is then classified using the previously learned causal decision boundary (and optionally checked by a VLM “boundary classifier” for hollow objects). This is the classification step.
>
> The “≈” signs (which will be replaced) indicate that the edited source tool is approximately equal, in the causally relevant feature space, to the novel object. In the figure two objects have the same length and tip angle after the first and second step respectively.
>
> >Typos. In Fig.2-left, the feature "suggestor" (which should be suggester?) is labelled with (e), whereas in the texts, it is labelled with . In Line 222, I did not find grasp points in Fig.3, so I assume it should be Fig.2?
>
> We thank you for catching these issues. All listed typos are fixed. In addition, we will perform a full pass for grammatical clarity and improve the figure references.

---

> > ### Author Response · Authors · 2025-11-18
> > **Response to Reviewer z4ew**
> >
> > >The authors use simulations with modified 3D objects to determine whether the suggested features are relevant to the task. Is it possible to replace this procedure with VLMs, which would be less time-consuming.
> >
> > Our pipeline uses the VLM to propose features but relies on simulation to perform feature‑isolated do-interventions [2] and determine which of those features are actually causal. Without this step, non‑causal features (which we see the VLM suggest in practice) would remain in the classifier and lead to systematic errors.
> >
> > We agree that simulation is more time‑consuming, but the causal discovery step is done offline once per task with the source tool before the robot is in the test environment. At deployment time, no simulation is required as the system classifies the tools by checking whether its inferred causal feature values fall within the working ranges established during simulation execution.
> > [2] Judea Pearl. Causality. Cambridge University Press, 2 edition, 2009.
> >
> >
> > >In Line 194, what are the 12 features in "top 6 most voted features out of 12"? Are they fixed for all tasks?
> >
> > The “12 features” are task-specific candidate features produced by our feature suggester VLM, not a fixed global set. For each task, we run the suggester ten times, aggregate all proposed features, and keep the 12 most frequently occurring ones. These constitute the candidate task-relevant features used both for (i) the human survey reported in Appendix A.7 and (ii) the initialization of our causal-reasoning pipeline, where we take the top six as the first intervention set. Appendix A.7 lists the full 12-feature sets for all three tasks.
> >
> > >With the relevant features, how does the classification step work? The texts in the paper are not very clearly written.
> >
> > Thank you for the clarification request. We will add the following text to our method:
> > This procedure is illustrated in Figure 2, right. We use the edit functions of each causal feature and make the source object as similar as possible to the novel object using the Chamfer distance for those features. Figure 2 shows that the source stick length is scaled to match the length of the novel object, ‘the selfie stick’. Then, its tip angle is changed to be the same as the selfie stick. This process continues for all object features and is repeated one more time to get the best match. The novel object is classified by checking whether its inferred causal feature values fall within the working ranges established during simulation execution. If the object is classified as suitable, we transfer the keypoints of the skill from the edited source tool to the novel object using Chamfer distance, and then execute the skill on the real robot. The limitations of this process for hollow objects are discussed in the experiments section (4.4); therefore, we also employ an additional classifier (VLM) for hollow objects.
> >
> >
> > >Sec 4.2 is titled "Task Performance and Inference Time Analysis", where is the inference time analysis? I only find one sentence with time-related information ("it requires minutes to scan a single object"), and it is not about inference.
> >
> > We agree with your assessment and will revise the text accordingly. We are in the process of measuring the time required to classify novel objects at test time and aim to incorporate these results into the revised version by next week. We will also rename the section to “Test Time Analysis” to better reflect its content.

---

### Official Review · Reviewer_VEAe · 2025-10-27

**Soundness:** 2
**Presentation:** 2
**Contribution:** 2
**Rating:** 2
**Confidence:** 4

**Summary:**

This paper is concerned with robots that can identify appropriate tools for their task at runtime, even if the available tools have never been seen before.  To address this challenge, the paper presents a computational pipeline that combines foundation models, procedurally generated object models, and physics simulators.  Given an existing tool known to be relevant to a task, new tools available in the environment are compared to the existing tool through several computational steps.  The steps include identification of relevant features on the existing tool, systematically warping each feature of the existing tool in a procedural object generator, simulating task execution with the warped versions to determine which work best, and then matching the best-performing warps to the novel tools in a new scene.  Foundation models and other deep models are used for several stages of this pipeline, including identification of relevant features, segmenting objects into parts, and reconstructing the current scene in a simulator.  Experimental results suggest that the proposed system has a higher success rate than several ablations and baselines.

**Strengths:**

- This paper works on an interesting problem and the high-level conceptual approach is appealing.  Simulation and procedural object warping are good ways to ground the error-prone recommendations of foundation models.
- The comparison with human-annotated object features is a nice touch.
- The work appears to be a substantial undertaking, given the combination of several large machine learning models and several robotic hardware platforms.

**Weaknesses:**

- One significant weakness concerns the technical novelty of the approach.  While some of the conceptual ideas may be novel, at the technical level, it appears that the computational pipeline does not involve any new method or model.  Rather, it combines several existing off-the-shelf foundation and other models in sequence.

- There are also deficiencies in the empirical analysis.  Table 1 presents overall success rate, but I did not see mention of the number of trials used to calculate success rate.  It is also not clear how much variation was present in each experimental condition (object poses, starting positions of the robots, etc.).  Several other "results" are only qualitative. In particular, lines 370-391 describe several trends anecdotally but do not provide any numerical data as quantitative empirical evidence for these claims.  And section 4.4 ("classification metrics") does not provide any metrics, it only references three images showing pictures of different objects.  Overall, there could have been a more comprehensive and intentional variation in experimental conditions, showing how performance varies in relation to those experimental conditions.  Details on computational expense would also be welcome, considering the large foundation models involved.

- Since the method uses foundation models trained on internet-scale data, the claim that "novel" tools have "never been seen before" is questionable.

- I saw several typos and grammatical errors; the presentation could be improved with more proof-reading.  Here are some:
    - section 3 line 127: missing period
    - page 4 line 189: "details if the controller" -> "details of the controller"?
    - page 4 line 203: "editor that gets a feature" > "editor gets a feature"?
    - page 5 line 222: "figure 3" should be "figure 2", right?
    - page 5 line 246-7: "that the all", "feautures"

**Questions:**

- It was unclear where the full feature set $F$ comes from.  If it is pre-defined per task, then can this method really be described as "open world"?

- If SAMPART3D requires a 3d object model as input, where does the 3d object model come from?  Is it the scene reconstruction?  But Fig 2 does not have a corresponding arrow from the scene reconstruction.

---

> ### Author Response · Authors · 2025-11-18
> **Response to Reviewer VEAe**
>
> We thank Reviewer VEAe for carefully engaging with our work and for their time. We are encouraged by the positive assessment of the problem setting and overall approach, the use of simulation and procedural object warping to ground foundation model suggestions, and the acknowledgement of both the human feature comparison and the effort across multiple models and robotic platforms. We answer their concerns and questions below:
>
> > One significant weakness concerns the technical novelty of the approach. While some of the conceptual ideas may be novel, at the technical level, it appears that the computational pipeline does not involve any new method or model. Rather, it combines several existing off-the-shelf foundation and other models in sequence.
>
> Our goal is to contribute a new capability rather than a new backbone model. Given a single reference tool and task, our system (i) identifies task-relevant semantic features of the tool via a VLM, (ii) turns these into parametric edits, (iii) performs do-style interventions[2] by procedurally warping each feature and evaluating the resulting tools in simulation, and (iv) uses the features whose interventions affect success as causal functional descriptors to classify and select novel target objects. Crucially, the feature space is not predefined for each task: candidate semantic features are proposed by the VLM and only become “real” once they are grounded through editing and physics. To the best of our knowledge, none of the related work offers this full pipeline of discovering semantic causal features, grounding them through controlled interventions in simulation, and then reusing them to select unseen physical tools.
>
> Imagine a robot serving candies by scooping them from a bowl. Unable to find a spoon, how can it find a replacement? Or how can it deduce that a cardboard tray can be used as well, without evaluating every object in the room (which would be infeasible)? This paper tries to frame and formalize this question, and it's here where the novelty lies. No paper to our knowledge formalizes this question. Of course, each component and their organization can further be improved, but the main focus of the paper is to show the feasibility of this approach.
>
> We also note that, in recent top venues such as ICLR 2025, several oral papers focus on new frameworks or capabilities built on existing models rather than on introducing new architectures. These include a probabilistic framework for LLM unlearning, optimal test-time compute allocation for LLM reasoning, and a large-scale analysis of data scaling laws in robot imitation learning. This pattern indicates that novelty in capability design, empirical understanding, and problem formulation is increasingly recognized as a substantive contribution in its own right.
>
> [2] Judea Pearl. Causality. Cambridge University Press, 2 edition, 2009.
>
> >There are also deficiencies in the empirical analysis.
>
> Thank you for the suggestions, we are running evaluations to improve this section. We will add them to openreview by next week.
>
> >Since the method uses foundation models trained on internet-scale data, the claim that "novel" tools have "never been seen before" is questionable.
>
> We appreciate this nuance and agree that, strictly speaking, we cannot guarantee that the underlying foundation models have never seen images of our tools on the internet. Our notion of “novel” is robot‑centric: the robot and our pipeline have no prior usage experience with these tools. In particular, the robot skill has never been executed with the novel tool, so the skill transfer must be fully formulated before any execution, and our pipeline did not (and cannot) run simulation rollouts with that specific tool to discover its task‑relevant causal features because it does not have a model of the tool. Causal features are found by only using the source tool. We will revise the wording in the paper to say that the tools are “novel to the robot and to our pipeline,” to more accurately reflect this intended meaning.
>
> >I saw several typos and grammatical errors; the presentation could be improved with more proof-reading.
>
> We thank the reviewer for catching these issues. All listed typos are fixed. In addition, we will perform a full pass for grammatical clarity.

---

> > ### Author Response · Authors · 2025-11-18
> > **Response to Reviewer VEAe**
> >
> > >It was unclear where the full feature set  comes from. If it is pre-defined per task, then can this method really be described as "open world"?
> >
> > The “full feature set” is not predefined per task. Conceptually, it is the (infinite) set of geometric and physical features that a tool could in principle have; our system only ever sees a subset of this space. In practice, for each task and reference tool we query a vision language model (VLM) to propose candidate features (e.g., “length of the handle,” “curvature of the spoon head”) that might be relevant for that task, and their causality are checked later by our pipeline. The features differ from task to task, the VLM generated features can be seen in Appendix (A.7). Therefore, we think our method is as open-world as a VLM is, which is as open-world as any existing AI technique. We will show proposed and causal features in the main text as well.
> >
> > >If SAMPART3D requires a 3d object model as input, where does the 3d object model come from? Is it the scene reconstruction? But Fig 2 does not have a corresponding arrow from the scene reconstruction.
> >
> > Thank you for the note, you are right that the current figure does not make this data flow explicit. For the source tool, SAMPART3D operates on the 3D model obtained from our scene reconstruction module [1]. It requires unoccluded multi-view images. Therefore, we only assume to access those for the source tool not for target tools. We will update the figure and text accordingly.
> >
> > [1] M. T. Villasevil, A. Simeonov, Z. Li, A. Chan, T. Chen, A. Gupta, and P. Agrawal, “Reconciling Reality through Simulation: A Real-To-Sim-to-Real Approach for Robust Manipulation,” in Proc. Robotics: Science and Systems, Delft, Netherlands, July 2024. doi: 10.15607/RSS.2024.XX.015.

---

### Official Review · Reviewer_SE88 · 2025-10-31

**Soundness:** 3
**Presentation:** 4
**Contribution:** 3
**Rating:** 6
**Confidence:** 4

**Summary:**

This paper addresses the challenge of creative robot tool use, enabling robots to repurpose or substitute everyday objects when the intended tool is unavailable. Existing approaches, such as affordance prediction or vision language reasoning, rely on correlation and semantic similarity rather than physical or causal understanding, which limits generalization to unseen tools. The authors propose a causal reasoning framework that discovers why a tool works by identifying its causally relevant physical features. A vision language model first proposes candidate features such as length, curvature, or mass. A shape editing module then generates counterfactual variants of the original tool by modifying these features, and a physics simulator evaluates each variant to determine which changes affect task success. The robot thereby learns a set of causal functional features that explain task performance. When encountering novel objects, the robot selects substitutes that share those causal properties. Experiments in simulation and real world manipulation tasks show that the system finds physically plausible substitutes, outperforms affordance and VLM based baselines, and produces interpretable causal explanations.

**Strengths:**

- Combines language model based hypothesis generation with physical counterfactual simulation.

- Produces interpretable explanations of which features make a tool functional.

- Enables successful substitution with previously unseen objects.

- Includes both simulation and real world experiments with clear gains over strong baselines.

**Weaknesses:**

- The causal analysis assumes high fidelity physics and accurate object reconstruction, which may reduce reliability in real world scenes.

- The evaluated tasks are limited in diversity and focus mainly on object centric scenarios; extending to deformable tools would strengthen the claim.

- Because candidate features originate from a vision language model, causally important but linguistically obscure attributes such as stiffness or friction may be missed.


- The related work section would benefit from citing closely related lines of research:

H. Chen, C. Zhu, S. Liu, Y. Li, and K. Driggs Campbell, “Tool as Interface: Learning Robot Policies from Observing Human Tool Use,” CoRL 2025.

M. Xu, P. Huang, W. Yu, S. Liu, X. Zhang, Y. Niu, T. Zhang, F. Xia, J. Tan, and D. Zhao, “Creative Robot Tool Use with Large Language Models,” arXiv:2310.13065, 2023.

**Questions:**

How robust is the causal feature discovery under perception or simulation noise

Can the method identify non linguistic causal attributes such as material compliance or friction

How transferable are discovered features across tasks, for example from scooping to pushing

How might the system generalize to multi step tool compositions or sequential creativity

---

> ### Author Response · Authors · 2025-11-18
> **Response to Reviewer SE88**
>
> We thank Reviewer SE88 for carefully engaging with our work and for their time. We appreciate the recognition of our combination of language model based hypothesis generation with physical counterfactual simulation, the interpretability of the discovered functional features, and the clear gains over baselines in both simulation and real world tool substitution experiments. We answer their concerns and questions below:
>
> >The causal analysis assumes high fidelity physics and accurate object reconstruction, which may reduce reliability in real world scenes.
>
> This is a good point and mentioned as a possible limitation in section 4.5. However, we should point out that the simulation is used only for the source object, and its parameters could be tuned while reconstructing the scene during  the offline causal discovery phase. We employ a state-of-the-art real-to-sim-to-real pipeline [1] which also leaves identifying physical parameters to future work because this can be hard without large interaction data (Section III.B of [1]). Therefore, we did not finetune the parameters either because it is not the primary concern of our method.
> That being said, we agree that there could be challenging objects to model, and explicitly analyzing robustness is valuable. We are currently working on adding controlled noise to simulation parameters to report performance changes in the revised version.
>
> [1] M. T. Villasevil, A. Simeonov, Z. Li, A. Chan, T. Chen, A. Gupta, and P. Agrawal, “Reconciling Reality through Simulation: A Real-To-Sim-to-Real Approach for Robust Manipulation,” in Proc. Robotics: Science and Systems, Delft, Netherlands, July 2024. doi: 10.15607/RSS.2024.XX.015.
>
> >The evaluated tasks are limited in diversity and focus mainly on object centric scenarios; extending to deformable tools would strengthen the claim.
>
> We appreciate this suggestion and agree that our current evaluation focuses on rigid everyday tools in the three tasks. We will soften the claims accordingly and consistently state that our contribution is a causal reasoning framework for rigid‑body tool use.
>
> Deformable tools require different modeling. Making a sponge longer could mean stretching it or extending it with the same deformation parameters. Theoretically, our pipeline can be extended to deformable objects with distinguishing assumptions like constant deformation parameters. However, our framework does not do that modeling in its current form and we will update our text accordingly.
>
> >Because candidate features originate from a vision language model, causally important but linguistically obscure attributes such as stiffness or friction may be missed.
>
> Although it could happen theoretically, we did not encounter this in practice for our tasks. The survey results in Appendix (A.7) show the alignment between human suggested features and VLM generated features. There, we see that VLM suggests friction in all tasks (no stiffness because probably source objects are rigid).
>
> To mitigate omissions, our pipeline is iterative: when a suggested tool fails in the real world, we assume that some causal features may be missing and re‑prompt the VLM for additional candidates, then rerun the counterfactual analysis (Sec. 3.2 and 4.5). In the hockey‑stick task, the initial iteration indeed overlooked mass, and the second iteration recovers it, allowing the system to reject overly heavy tools such as metal crowbars.
>
> If a truly crucial attribute cannot be articulated by the VLM after several iterations, our current system cannot discover it. For the everyday tools and tasks we evaluate, we did not encounter such failure cases, though we acknowledge that they could arise, as noted in Sec. 4.5. More broadly, discovering arbitrarily useful physical characteristics is inherently challenging; however, we believe that our framework offers the strongest open-world approach to date by grounding proposed features through physical simulation.
>
> >‘The related work section would benefit from citing closely related lines of research:
> H. Chen, C. Zhu, S. Liu, Y. Li, and K. Driggs Campbell, “Tool as Interface: Learning Robot Policies from Observing Human Tool Use,” CoRL 2025.
> M. Xu, P. Huang, W. Yu, S. Liu, X. Zhang, Y. Niu, T. Zhang, F. Xia, J. Tan, and D. Zhao, “Creative Robot Tool Use with Large Language Models,” arXiv:2310.13065, 2023.’
>
> Our current related work section already discusses RoboTool in line 112. We are also adding the recent work “Tool as Interface” by acknowledging that they also use object reconstruction to learn tool use policies from human videos in the related work.

---

> > ### Author Response · Authors · 2025-11-18
> > **Response to Reviewer SE88**
> >
> > >How robust is the causal feature discovery under perception or simulation noise?
> >
> > We agree that it is important to quantify this intuition. Although our real world experiments include perception and motor noise, we are working to add robustness analysis to the rebuttal by next week.
> >
> > >Can the method identify non linguistic causal attributes such as material compliance or friction?
> >
> > Discovery in simulation: Our feature suggester could come up with such properties like compliance and friction as described above, and our simulators allow us to edit this property to discover causal relevance of the features to the task. In the experiments, we already treat mass in exactly this way: mass is proposed by the VLM, we edit it directly in the simulator.
> >
> > Discovery in the real world: However, using inferred features to classify novel tools in the real world requires estimating values of the features, and it is more involved than estimating mass. Mass can be directly inferred from the torque values while lifting the tool, but properties like compliance and friction would require some exploratory actions for each object which would be costly. We frame our work as fast adaptation to novel tools, therefore, we did not include such actions as a design choice.
> >
> > In short, we can identify causal relevance of physical properties as long as they are suggested by VLMs, the simulator supports them, and inference in the real world does not require complex interactions.
> >
> > >How transferable are discovered features across tasks, for example from scooping to pushing?
> >
> > Our pipeline intentionally discovers causal features dynamically per task: we run the feature suggester and counterfactual experiments from scratch for each new task, and the resulting feature set can differ even for the same physical object. So transferring features to new objects in new tasks is not in the scope of the paper. We are aiming to find task-specific features for objects that would change between tasks. We illustrate this in pushing and scooping tasks. Although shepherd cane and yoga stick are included in both tasks, they are evaluated for different features.
> >
> > >How might the system generalize to multi step tool compositions or sequential creativity?
> >
> > This is an important next step, and it is the focus of our upcoming work. We believe that achieving multi-step tool composition and sequential creativity requires a hierarchical planning mechanism. In such a setup, our current framework would function as a tool-selection and tool-usage oracle for creativity at the level of individual subgoals. However, there is an additional layer of creativity required at the level of high-level plan generation. Addressing that challenge is a separate problem, which is why it falls outside the scope of the present work and is left for future research.

---

### Author Response · Authors · 2025-11-18
**Response to all reviewers**

We thank all reviewers for their time, careful reading, and constructive feedback on our submission. We are happy to see that reviewers found the problem that we are tackling important. Below, we address the questions seeking clarification point by point. For comments that require additional experiments or further evaluation, we will incorporate next week the corresponding results and analyses both in the rebuttal and in the revised version of the paper.

To sum up, while there is certainly room to improve individual components of the system, our primary contribution is **to formulate and demonstrate a new capability**. Imagine a robot serving candy by scooping them from a bowl. If the robot cannot find a spoon, how can it find a replacement, or how can it deduce that a cardboard tray can be used as well and transfer its skill to the cardboard tool? Given a single reference tool and task, our system (i) identifies task-relevant semantic features of the tool via a VLM, (ii) turns these into parametric edits, (iii) performs **do-style interventions** (Pearl, The Book of Why) by procedurally warping each feature and evaluating the resulting tools in simulation, and (iv) uses the features whose interventions affect success as **causal functional descriptors** to classify and select novel target objects. Crucially, the feature space is **not predefined for each task**: candidate semantic features are proposed by the VLM and **only become “real” once they are grounded through editing and physics**. To the best of our knowledge, **none of the related work offers such a pipeline** of discovering semantic causal features, grounding them through controlled interventions in simulation, and then reusing them to select unseen physical tools, meaning the robot has never **used or interacted with them before**.

---

### Meta-Review · Area_Chair_DX7U · 2026-01-01

**Summary:**

This paper provides a compelling demonstration of how to use VLMs in tandem with physics simulators to allow robots to select new tools to use that they have never seen before. However, there are several limitations as noted by the reviewers:

### Limited technical novelty
All reviewers mentioned limited technical novelty, as the system is a combination of existing techniques to allow a new capability. However, I believe this is generally not a reason for rejection, as long as the resulting pipeline is thoroughly investigated.

### Deficient empirical analysis
All reviewers noted deficiencies in the empirical analysis of the pipeline, either because very few tasks were tested, or because of specific missing tests (e.g. for generalizability, robustness to noise, or inference time).

### Limited tasks
2 reviewers point out that only 3 tasks are assessed, and many of the rebuttal points by the authors note that they simply did not see issues within the limited tasks they assess here. Given the potential fragility of the approach, and the lack of empirical analysis, having very few tasks raises the concern that this pipeline is essentially "cherry-picked" for these particular settings.

Despite their promise, the authors *did not upload a revised version of the manuscript with the recommended changes from reviewers as promised*, which included critical analyses and improvements to the writing. Therefore, I recommend rejection.

**Reviewer Concerns:**

### Limited technical novelty - addressed
Many reviewers mentioned concerns that the paper simply pulls together existing tools into a new framework. I agree with the authors that this is a legitimate contribution nonetheless, since there is a new capability that is enabled.

### Limited analysis - unaddressed
However, in such a scenario (where the main contribution is a pipeline), and as noted by reviewers, it is extra important to conduct a thorough empirical analysis to understand how and where those components fail for the new capability. Despite several promises by the authors to run the extra analyses, none of these were run for the rebuttal. Given the long rebuttal period, and the importance of some of these experiments (e.g. inference time analysis), this was not a good faith effort from the authors to satisfy reviewer concerns.

**Reviewer Scores:**

Reviewer SE88
* I do not believe this reviewer would have changed their score. They asked for an evaluation of the robustness to noise. The authors promised "we are working to add robustness analysis to the rebuttal by next week." but they did not add these results to the rebuttal.

Reviewer VEAe
* I do not believe this reviewer would have changed their score. They asked for a more detailed empirical analysis, which the authors promised "Thank you for the suggestions, we are running evaluations to improve this section. We will add them to openreview by next week." but these were not added.

Reviewer z4ew
* I do not believe this reviewer would have changed their score. They asked for an inference time analysis. The authors promised: "We are in the process of measuring the time required to classify novel objects at test time and aim to incorporate these results into the revised version by next week." but this was not added.

---

### Decision · Program_Chairs · 2026-01-26

Reject